# Metabolomic Analysis Reveals Changes in Preimplantation Embryos Following Fresh or Vitrified Transfer

**DOI:** 10.3390/ijms21197116

**Published:** 2020-09-26

**Authors:** Ximo Garcia-Dominguez, Gianfranco Diretto, Sarah Frusciante, José Salvador Vicente, Francisco Marco-Jiménez

**Affiliations:** 1Laboratory of Biotechnology of Reproduction, Institute for Animal Science and Technology (ICTA), Universitat Politècnica de València, 46022 Valencia, Spain; ximo.garciadominguez@gmail.com (X.G.-D.); jvicent@dca.upv.es (J.S.V.); 2Italian National Agency for New Technologies, Energy and Sustainable Development (ENEA), Casaccia Research Centre, 00123 Rome, Italy; gianfranco.diretto@enea.it (G.D.); sarah.frusciante@enea.it (S.F.)

**Keywords:** embryo manipulation, cryopreservation, stress, metabolism, developmental plasticity, developmental programming

## Abstract

Although assisted reproduction technologies (ARTs) are recognised as safe, and most of the offspring seem apparently healthy, there is clear evidence that ARTs are associated with changes in the embryo’s developmental trajectory, which incur physiological consequences during the prenatal and postnatal stages of life. The present study aimed to address the influence of early (day-3 embryos) embryo transfer and cryopreservation on embryo survival, size, and metabolome at the preimplantation stage (day-6 embryos). To this end, fresh-transferred (FT) and vitrified-transferred (VT) embryos were compared using naturally-conceived (NC) embryos as a control reference. The results show that as in vitro manipulation was increased (NC < FT < VT), both embryo survival rate (0.91 ± 0.02, 0.78 ± 0.05 and 0.63 ± 0.05, for NC, FT, and VT groups, respectively) and embryo size (3.21 ± 0.49 mm, 2.15 ± 0.51 mm, 1.76 ± 0.46 mm of diameter for NC, FT, and VT groups, respectively) were significantly decreased. Moreover, an unbiased metabolomics analysis showed overall down-accumulation in 40 metabolites among the three experimental groups, with embryo transfer and embryo cryopreservation procedures both exerting a cumulative effect. In this regard, targeted metabolomics findings revealed a significant reduction in some metabolites involved in metabolic pathways, such as the Krebs cycle, amino acids, unsaturated fatty acids, and arachidonic acid metabolisms. Altogether, these findings highlight a synergistic effect between the embryo transfer and vitrification procedures in preimplantation embryos. However, the ex vivo manipulation during embryo transfer seemed to be the major trigger of the embryonic changes, as the deviations added by the vitrification process were relatively smaller.

## 1. Introduction

During early embryonic development, the maternal environment supports optimal embryogenesis, providing stage-specific nutrients, growth factors, hormones, antioxidants, and other regulatory molecules that are required by the embryo’s evolving developmental and metabolic requirements [1,2,3,4]. Mounting evidence suggests that early exposures to stressful situations that affect these elegantly orchestrated milestones may influence developmental trajectories in the short and long term [5,6]. Under these suboptimal situations, the embryo’s developmental plasticity is adept at compensating this environmental drawback if the stress is within the normal range [7]. This mechanism may facilitate the setting of suitable growth and metabolic parameters to match the perceived environment but, if environmental conditions change later, developed adaptations may become maladaptive [5]. Besides, extreme exposures outside the normal range are associated with abnormal widespread epigenetic remodelling, which can persist through subsequent cell cycles and drive an altered developmental programme [5,6,7]. Illuminating pieces of evidence come from the fields applying Assisted Reproductive Technologies (ARTs), which expose gametes and the early embryo to precisely timed in vitro manipulations in static culture mediums, under some physical and chemical stressors, and deprived of the maternal interface [4,8]. Thus, being conscious of the limitations when recapitulating the in vivo dynamism, and although the majority of ART-offspring show healthy development, safety concerns regarding ARTs are today a matter of debate [9]. Extensive reviews have reported that ART-stressors during in vitro fertilisation (IVF) and in vitro culture (IVC) may not only impair embryo development in the laboratory, but can also have downstream effects after the embryo transfer across gestation and after birth [7,10,11,12,13]. One potential strategy to attenuate the suboptimal conditions is the use of oviductal and uterine secretions and/or cells, which results in embryos closer to those conceived naturally [14,15,16,17]. As the increasing use of ARTs continues to spread worldwide, mimicking in vivo conditions is especially important to guarantee the safety of ART procedures, but some of them are extremely contradictory with this concept, such as embryo cryopreservation.

Regardless of the cryopreservation method used, all of them require embryos to be exposed to an environment in which they have no intrinsic ability to survive (i.e., toxic cryoprotectant solutions and ultra-low temperatures [18]). The study of the embryo cryopreservation legacy in the developmental trajectory in utero and throughout life has merited our attention for decades. In this sense, we proved that embryo cryopreservation disturbs preimplantation embryo gene expression, increases foetal losses and placental remodelling, modifies both prenatal and postnatal growth trajectories, and incurs phenotypic changes in adulthood [19,20,21,22,23,24,25,26]. However, during a vitrified embryo transfer procedure, embryos must necessarily be recovered, catalogued, vitrified, warmed, and transferred, so the role of individual stressors has not currently been differentiated, and all of them are considered together as part of the whole process. Particularly, Garcia-Dominguez et al. [26] have shown that both embryo transfer and embryo vitrification procedures are important triggers of long-term developmental reshaping. Therefore, the importance of each stressor and its interactions (additive, antagonistic or synergistic effects) during an ART-cycle in the embryo trajectory should not be underestimated. In this scenario, the aim of this study was to address the influence of in vitro manipulation during fresh and vitrified embryo transfer procedures on the preimplantation embryo development and metabolome.

## 2. Results

### 2.1. In Vivo Embryo Development

Data showed that the survival rate decreased as embryonic manipulation increased, noting lower values for vitrified-transferred (VT) embryos (0.63 ± 0.05) than for fresh-transferred (FT) embryos (0.78 ± 0.05; *p* = 0.042). Both VT and FT presented lower survival rates than naturally-conceived (NC) embryos (0.91 ± 0.02; *p* = 0.000 and *p* = 0.017 for NC vs. VT and NC vs. FT comparison, respectively). The foster mother effect was not significant and the estimate effect was 0.29 ± 0.25; *p* = 0.0239. The morphometric details of day-6 in vivo developed embryos were noted in Table 1, showing that embryos were also smaller as in vitro manipulation increased. Then VT embryos presented a lower diameter area and perimeter than FT embryos, whose size was lower compared to the NC group.

### 2.2. Comparative Study of the Metabolome between In Vivo Developed Embryos

The metabolomic profiles of VT, FT, and NC in vivo developed embryos were compared in order to unravel the effects attributable to the embryo transfer procedure (FT vs. NC), to the vitrification procedure per se (VT vs. FT) and to the whole vitrified embryo transfer procedure (VT vs. NC). First of all, we carried out an untargeted metabolomics analysis of both semi-polar and non-polar fractions to gain a general overview of the metabolic changes occurring. The t-test analysis detected 40 differentially accumulated metabolites (DAMs) for at least one of the above comparisons, however, no significant associations were observed (false discovery rate, FDR > 0.05). Of these, 29 belonged to the semi-polar fraction and 11 to the non-polar. Principal components analysis (PCA) biplots demonstrated clustering among the three groups using the 87 and 27 metabolites detected in semi-polar and the non-polar fractions, respectively (Figure 1A,B). 

Focusing on the semi-polar fraction (Figure 1A), distant clusters were observed for NC and VT embryos, the FT group being intermediate, as expected. Differences among the non-polar metabolite profiles appeared more elusive (Figure 1B), although they seemed to follow the same trend as above, the FT cluster being allocated between those of NC and VT. This demonstrated that deviations caused during a vitrified embryo transfer (VT/NC) were higher than those induced after a fresh embryo transfer (FT/NC), as vitrification procedure (VT/FT) had effects per se that were additive to the ex vivo manipulation. Supporting this evidence, general down-accumulation was recorded after fresh embryo transfer, vitrified embryo transfer and embryo vitrification procedure, the average fold change being −0.62, −0.81, and −0.20 for semi-polar metabolites, and −2.64, −3.42, and −1.28 for the non-polar ones, respectively. More specifically, Venn diagrams of the semi-polar (Figure 1C) and non-polar (Figure 1D) DAMs showed that, although each procedure induced some condition-specific effects, most of the DAMs that appeared after the vitrified embryo transfer can be attributed to the embryo transfer procedure, being less attributable to the vitrification process. Particularly, of the total DAMs noted after the vitrified embryo transfer, 46.4% (13/28) and 28.6% (8/28) of them were influenced by the embryo transfer and embryo vitrification, respectively. Notably, 39.3% (11/28) was not directly attributable to either procedure, suggesting that a coexistence of both triggers resulted in a synergistic effect with long lasting metabolic alterations.

After that, a targeted analysis was performed to quantify 74 known metabolites, in a relative way, of the primary and secondary metabolisms (Appendix A). The t-test showed 15 DAMs (*p* < 0.05) from which none of them had significant FDR (Table 2). Functional clustering of the metabolites revealed alterations in metabolic pathways related to the Krebs cycle, amino acids metabolism, unsaturated fatty acid metabolism and arachidonic acid metabolism. In concordance with the untargeted data, general down-accumulation was recorded for all the comparisons after targeted analysis. Altogether, metabolomics data revealed that ex vivo manipulations during the embryo transfer technique exerted a greater impact over the preimplantation embryo metabolism than the embryo vitrification procedure. Thus, the main differences observed after a vitrified embryo transfer procedure seemed to be caused by the embryo transfer procedure.

## 3. Discussions

Throughout this study, it has been demonstrated that more severe in vitro stress precipitated more deviant phenotypes and metabolic alterations on preimplantation embryos. Moreover, to our best knowledge, metabolomics was here used for the first time to address the influence of the embryo transfer and vitrification procedures on the embryonic metabolite content. Although both techniques are significant triggers of specific and cumulative effects, strong evidence of the negative synergy between both stressors was provided. However, ex vivo manipulations, rather than embryo vitrification, seemed to be largely responsible for both phenotypic and molecular deviations.

The results shown here indicate that, as embryo manipulation increased, early embryos seemed to lose its developmental capacity within the first 3 days. Knowledge so far indicates that suboptimal conditions during ART may induce embryonic stress responses and affect embryo viability [10,11]. In addition, cryopreservation is known for its harmful impacts on embryonic health, placing embryos at risk from a variety of types of cryoinjury during temperature and phase transitions [18,27]. Such survival impairment is in agreement with previous studies performed in rabbits, where manipulations of either fresh or vitrified embryos resulted in negative developmental effects both under in vitro conditions and in vivo [22,23,26,28,29,30]. It has been reported that successful hatching is dependent on a sufficiently high number of embryonic cells, which enables blastocyst expansion and zona shedding [31]. However, as more in vitro stress was applied, the number of total embryonic cells became more reduced [32], embryo cryopreservation being a significant cause of cell loss [33]. This could explain both the smaller embryo size and the lower embryo survival of VT embryos compared to those FT, the values for the NC group being higher than for the two previous ones. To date, ART procedures have been linked with a lower blastocyst development, slower developmental kinetics, reduced number of cells, higher incidence of apoptosis, and great dysregulations in the epigenetic, transcriptomic, and metabolic cell profiles [10,11]. Then, identifying the molecular changes occurring in developing embryos following ART procedures could unveil key mechanisms to better understand why these procedures generate embryos with altered developmental trajectories.

In the present study, 40 endogenous DAMs, globally down-accumulated, were identified among VT, FT, and NC embryos though an unbiased (untargeted) metabolomic approach. The study of these DAMs through PCA and Venn diagrams suggested that both the transfer technique and the vitrification procedure induced significant modifications in the metabolic content in preimplantation embryos. In agreement, previously reported studies confirmed that both stressors altered gene expression in rabbit embryos and increased the embryonic loses [22,30]. Besides, we observed that ex vivo manipulations during the embryo transfer procedure, rather than embryo vitrification, seemed to induce the major metabolic change following vitrified embryo transfer. The study conducted by Salilew-Wondim et al. revealed that the shortest in vitro exposure, compared to development entirely in vivo, induced notable changes in the embryonic epigenetic profile, the changes induced by longer in vitro culture times being less prominent [34]. Taking these findings into account, we hypothesise that embryonic responses are highly sensitive to the stress conditions, as disturbing optimal maternal environment by in vitro embryo handling seems to incur higher effects compared to those induced by apparently more stressful ARTs such as embryo cryopreservation. In addition, there are significant elements that add to the complexity of these phenomena, as several ART-related developmental effects could be synergic, condition-specific, strain-specific, sexually dimorphic, or may not emerge until later into adulthood [11,13]. Here, we proved that the coexistence of two stressors (i.e., ex vivo manipulation and vitrification) induced a synergic effect that incurred more metabolic changes than each one individually. Details of the phenotypic variation following both fresh and vitrified embryo transfer at adulthood were described in our recent study, detecting a cumulative effect between both procedures [26].

To more clearly interrogate the metabolic changes associated with each stressor, we performed a targeted analysis, identifying an overall down-accumulation in pivotal metabolites involved in the Krebs cycle, amino acid metabolism and unsaturated fatty acid metabolism. These results were in agreement with those found in bovine and porcine after ART at the gene expression level [35,36,37,38,39]. Particularly, Swain et al. showed that in vitro produced pig embryos had reduced metabolic activity, mainly related with glycolytic pathways and the Krebs cycle, and used lesser amounts of glutamine, compared with in vivo derived embryos at all stages [40]. Concordantly, we observed an overall negative fold changes and low amounts of succinate and glutamine in both FT and VT embryos. Besides, other studies in the mouse reported changes in the amino acid utilisation after in vitro embryo production, which has been importantly linked with the embryo developmental kinetics [41,42]. Particularly, a bovine study demonstrated that fast-growing embryos consumed higher threonine amounts than the rest [43]. Supporting this evidence, the present study revealed that both FT and VT embryos were smaller and exhibited lower threonine content compared to the NC ones. Importantly, some n-3 and n-6 polyunsaturated fatty acids (i.e., eicosapentaenoic acid, docosapentaenoic acid, adrenic acid, and arachidonic acid) were also found down-accumulated in both FT and VT embryos. These compounds mediate the regulation of genes involved in lipid and carbohydrate metabolism, as well as the lipogenic pathways related to the fatty acid, triglycerides, and cholesterol synthesis [44]. Therefore, dysregulations of these metabolites could exert some downstream alterations that can contribute to the metabolic and developmental changes following fresh or vitrified embryo transfer. In particular, arachidonic acid is a precursor for the synthesis of prostaglandins, which are vital for the embryo-maternal dialogue, for the shedding of the zona pellucida and, thereby, for the subsequent embryo survival and development [44,45]. As some metabolites stemmed from the metabolism of arachidonic acid were here detected down-accumulated in FT and VT embryos, it could explain its lower survival rates compared to the NC ones. It is worth noting that some of the metabolic changes here described have also been reported in adult tissues of ART animals [13,46,47,48]. Therefore, the study of the molecular changes occurring in the preimplantation embryo after ART could provide a set of compounds that can serve as metabolic markers to better understand and track those deviations that appear both in the short and in the long term. In accordance with the “quiet range” of metabolic activity, embryos with maximum developmental potential will be located in a “Goldilocks zone”, understood as an optimal range of metabolic activity and energetic efficiency [49]. Then, following this idea, only those embryos able to adapt their metabolism within a homeostatic range during suboptimal ART condition could finally develop until birth. In agreement, current developmental biology has begun to describe different ways whereby the metabolic profile of the cell and developmental programmes of the organism can crosstalk, establishing the notion that metabolism is not just a housekeeping process, but instead an active effector of physiological changes [50]. In this context, the Developmental Origins of Health and Disease (DOHaD) hypothesis evokes the idea that a wide range of conditions present in adulthood have their origin in these embryonic adaptations [10,11,12]. Given the increasing numbers of ART cycles that have been performed annually, both in humans and in animal production, monitoring the ART legacy in both embryos and adult individuals should be mandatory and of special interest, in order to ensure that ARTs are used in the safest possible way.

In summary, here we provide strong evidence of the metabolic reprogramming undergone by in vivo developed preimplantational embryos following both fresh and vitrified embryo transfer. We have demonstrated that both embryo transfer and embryo cryopreservation are significant stressors triggering developmental reshaping, whose coexistence can induce a synergic effect causing more deviant alterations. Additional research is required to elucidate whether the embryonic metabolic changes against suboptimal in vitro condition can persist in the course of development and can be used as predictors of embryo quality and adult health.

## 4. Materials and Methods

Unless stated otherwise, all chemicals were purchased from Sigma-Aldrich Corporation (St Louis, MO, USA).

### 4.1. Animals and Ethical Statements

New Zealand rabbits belonging to the Universitat Politècnica de València were used throughout the experiment. The animal study protocol was reviewed and approved (code: 2015/VSC/PEA/00061) by the “Universitat Politècnica de València” ethical committee prior to initiation of the study. All experiments were performed in accordance with guidelines and regulations set forth in Directive 2010/63/EU EEC. Animal experiments were conducted at an accredited animal care facility (code: ES462500001091).

### 4.2. Experimental Design

Figure 2 illustrates the experimental design, conceived to elucidate the effects of transferring either fresh or vitrified-warmed embryos on their preimplantation development and metabolism. NC embryos were used as control reference. With this aim, a total of 12 donor females were induced to superovulate using 3 μg of corifollitropin alpha [51]. After 3 days, females were inseminated with a semen pool from unrelated males with proven fertility, ovulation being induced by an intramuscular injection of 1 µg of buserelin acetate (Hoechst Marion Roussel, Madrid, Spain). Three days post-insemination, a total of 263 embryos catalogued as normal (presenting homogenous cellular mass and spherical mucin coat and zona pellucida) were recovered post-mortem from 7 females. All embryos were pooled for later distribution in the different parts of the study, thus reducing the effect of embryo donors. Of the total, 150 embryos were subjected to vitrification, keeping 148 undamaged embryos (presenting homogenous cellular mass, mucin coat, and spherical zona pellucida) after warming. Then, 148 warmed and 113 fresh embryos were oviductally transferred into 7 foster mothers (35–40 embryos per foster mother). After 3 days (day 6 of embryonic development), 93 vitrified-transferred (VT group) and 88 fresh-transferred (FT group) preimplantation embryos were recovered. In addition, from the remaining 5 inseminated females, 176 coetaneous 6-day preimplantation embryos were obtained without embryo manipulation (naturally-conceived group; NC). The survival rate (recovered embryos/transferred embryos) was noted, and digital images were obtained to perform a morphometric study. Finally, 8 pools of 10 in vivo developed preimplantation embryos were generated for each experimental group (NC, FT, and VT), which were used to perform a comparative metabolomics study.

### 4.3. Embryo Vitrification

Vitrification was achieved in two steps, following the protocol adapted from previous studies [52,53,54]. Briefly, in the first step, embryos were placed for 2 min in a solution consisting of 10% (*v*/*v*) dimethyl sulfoxide (DMSO) and 10% (*v*/*v*) ethylene glycol (EG) in base medium. Base medium (BM) consisted of Dulbecco’s phosphate buffered saline supplemented with 0.2% (*w*/*v*) of bovine serum albumin. In the second step, embryos were suspended for 1 min in a solution of 20% DMSO and 20% EG in BM. Then, embryos were loaded into Cryotop® devices (8 embryos per device), the surrounding vitrification solution was removed, and devices were rapidly plunged into liquid nitrogen to achieve vitrification. For warming, embryos were placed in 2 mL of 0.33 M sucrose in BM at 25 °C to remove cryoprotectants, and washed 5 min later with BM.

### 4.4. Embryo Transfer

Warmed or fresh embryos were laparoscopically transferred into the oviduct of foster mothers, following the protocol described by Besenfelder and Brem [55]. Briefly, foster mothers were anaesthetised with xylazine (5 mg/kg; Rompun; Bayern AG, Leverkusen, Germany) intramuscularly and ketamine hydrochloride (35 mg/kg; Imalgene 1000; Merial S.A.S., Lyon, France) intravenously, and placed in Trendelenburg’s position. Then, embryos were loaded in a 17G epidural catheter, which was inserted through a 17G epidural needle into the inguinal region. Finally, while the process was monitored by single-port laparoscopy, the catheter was introduced into the oviduct through the infundibulum to release the embryos. Using this procedure, between 35 and 40 embryos were transferred to each pseudopregnant foster mother induced to ovulate 68–72 h before transfer. Both embryo vitrification and transfer processes used in this experiment were described in detail in Garcia-Dominguez et al. [54].

### 4.5. In Vivo Embryo Development

A total of 261 embryos (148 vitrified-warmed and 113 fresh) were transferred into 7 foster mothers and in vivo cultured during 72 h. Then, foster mothers and 5 pregnant females were euthanised, and day-6 embryos were recovered by perfusion of each uterine horn with 10 mL of pre-warmed BM. The survival rate was noted taking into account the number of embryos transferred and those recovered. In the case of the NC group, the number of available embryos was estimated from the number of corpora lutea (ovulation rate) in the ovaries, as has been used before [26].

### 4.6. Morphometric Analysis

In vivo developed embryos were subjected to a morphometric analysis to compare the embryo size between the experimental groups (VT vs. FT vs. NC). All embryos were photographed and measured (diameter, perimeter, and area) in digital images using ImageJ analysis software (http://rsb.info.nih.gov/ij/).

### 4.7. Semi-Polar and Non-Polar Metabolome Analysis

In vivo developed embryos were subjected to a comparative metabolomics analysis. For each experimental group, 8 independent biological replicates, consisting of 10 embryos each, were analysed. Then, 12 pools (4 VT, 4 FT and 4 NC) were used to study the semi-polar fraction, and 12 pools (4 VT, 4 FT and 4 NC) were used to study the non-polar one. For each biological replicate, at least one technical replicate was carried out. Targeted and untargeted liquid chromatography-electrospray ionisation-high resolution mass spectrometry (LC-ESI-HRMS) analysis of the semi-polar metabolome were performed as previously described [56,57,58], while targeted and untargeted liquid chromatography-atmospheric pressure chemical ionisation-high resolution mass spectrometry (LC-APCI-HRMS) analyses of the non-polar metabolome were carried out as reported previously [59,60,61]. Untargeted metabolomics was performed using the SIEVE software (Thermo Fisher scientific). Principal components analysis biplot (PCA-biplot) of untargeted metabolomes were created with BioVinci software (BioTuring Inc., San Diego, CA, USA). Targeted metabolite identification was performed by comparing chromatographic and spectral properties with authentic standards (if available) and reference spectra, in-house database, literature data, and on the basis of the m/z accurate masses, as reported in the PubChem database (http://pubchem.ncbi.nlm.nih.gov/) for monoisotopic mass identification, or on the Metabolomics Fiehn Lab Mass Spectrometry Adduct Calculator (http://fiehnlab.ucdavis.edu/staff/kind/Metabolomics/MS-Adduct-Calculator/) in the case of adduction detection. Finally, DAMs were achieved as described above. Metabolites were quantified in a relative way by normalisation on the internal standard (formononetin and DL-α-tocopherol acetate) amounts.

### 4.8. Statistical Analysis

#### 4.8.1. Survival Rate and Embryo Morphology Statistical Analysis

Differences in the survival rate and embryo diameter, perimeter, and area were assessed applying a general linear model described above:Y_ij_ = μ + P_i_ + CO_j_ + e_ij_
where Y_ij_ was the trait to analyse, µ was the general mean, P_i_ was the fixed effect of experimental groups (VT, FT, and NC), CO_j_ was the random effect of foster mother and e_ij_ was the error term of the model. The random effect donor mother (CO_j_). Survival rate was evaluated using a function probit link with binomial error distribution. A *p*-value < 0.05 was considered indicative of a statistically significant difference. The data are presented as mean ± standard error. Statistical analyses were performed with the SPSS 21.0 software package (SPSS Inc., Chicago, IL, USA).

#### 4.8.2. Metabolome Statistical Analysis

After chromatogram alignment and retrieval of all of the detected frames (e.g., ions), differentially accumulated metabolites (DAMs) were validated by a statistical analysis (one-way ANOVA plus Tukey’s pairwise comparison) using the SPSS software (SPSS Inc., Chicago, IL, USA). Benjamini–Hochberg false discovery rate (FDR) was performed to correct for multiple testing (*p*  >  0.05) [62].

## 5. Limitations

The number of included samples is relatively small, which may restrict the interpretation of our results. Thus, low sample size negatively affects the likelihood that a nominally statistically significant finding reflects a true effect. Our results are based on *p*-value without correct for multiple comparisons. The *p*-values have been listed for every comparison. Further study needs to validate our results in a larger study sample. No other embryo develop time-points were examined. Therefore, temporal changes in metabolites profiles during embryogenesis remain elusive. The choice to analyse preimplantation rabbit embryos was based on our previous findings [19]. Consequently, the potential involvement of ex vivo manipulation during assisted reproduction technologies of our animal model remains to be determined. This model is characterised by the long-term and transgenerational phenotypic, transcriptional, and metabolic effects in rabbit [63].

## Figures and Tables

**Figure 1 ijms-21-07116-f001:**
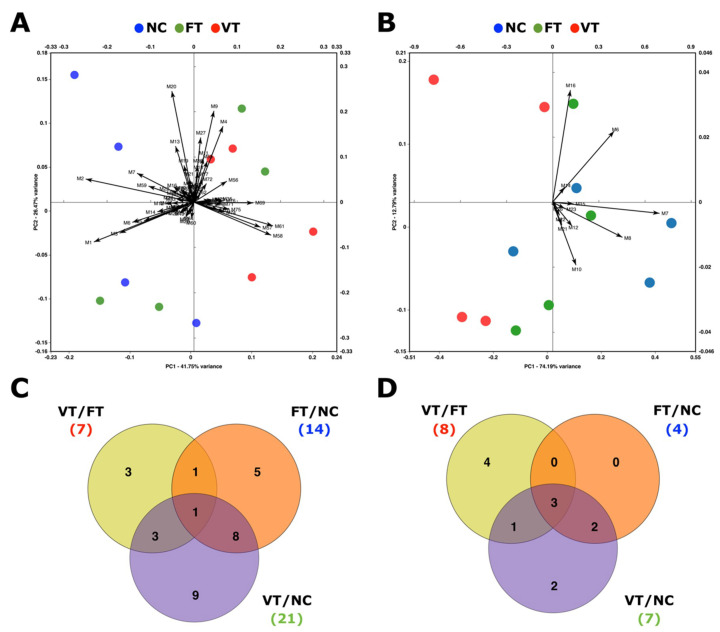
Unbiased metabolomic data (untargeted study) comparing the metabolic profile of naturally-conceived (NC) embryos and those in vivo developed after fresh embryo transfer (FT) and vitrified embryo transfer (VT). Principal components analysis biplots of the (**A**) semi-polar and (**B**) non-polar metabolites. Venn diagrams comparing the number of targeted (**C**) semi-polar and (**D**) non-polar differentially accumulated metabolites after fresh embryo transfer (FT/NC), vitrified embryo transfer (VT/NC), and embryo cryopreservation (VT/FT).

**Figure 2 ijms-21-07116-f002:**
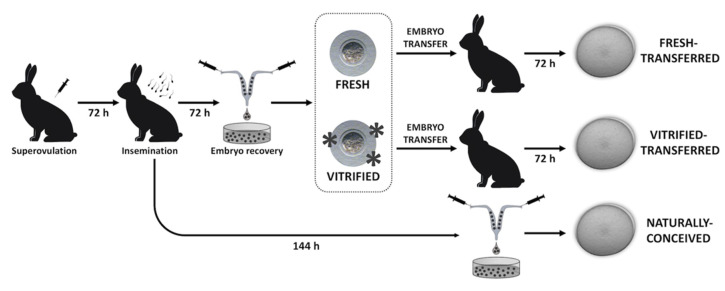
Experimental design. First of all, day-5 fresh and vitrified embryos were compared after 48 h of in vitro development under standard conditions. Their developmental potential and morphometric features were assessed. Moreover, day-6 preimplantation embryos obtained after fresh and vitrified embryo transfer were compared using naturally conceived embryos as reference. In this case, their recovery rates, morphometric features, and metabolomics profiles were analysed.

**Table 1 ijms-21-07116-t001:** Morphometric analysis of 6-day-old embryos after in vivo development by natural conception or following fresh or vitrified embryo transfer.

Experimental groups	n	Diameter (mm)	Perimeter (mm)	Area (mm^2^)
Vitrified-transferred	93	1.76 ± 0.46 ^c^	5.81 ± 1.55 ^c^	2.63 ± 1.34 ^c^
Fresh-transferred	88	2.15 ± 0.51 ^b^	7.27 ± 1.67 ^b^	3.79 ± 1.67 ^b^
Naturally-conceived	176	3.21 ± 0.49 ^a^	10.67 ± 1.47 ^a^	8.15 ± 2.39 ^a^
*p*-value		0.005	0.003	0.002
Foster mother effect	7	0.04 ± 0.03	0.36 ± 0.31	0.35 ± 0.36
*p*-value		0.249	0.241	0.337

n: number of embryos. Data are expressed as mean ± standard error. ^a,b,c^ Values within a column with different superscripts differ statistically.

**Table 2 ijms-21-07116-t002:** Targeted identification of differentially accumulated metabolites in day-6 embryos after in vivo development by natural conception or following fresh or vitrified embryo transfer. Asterisk denotes statistical differences. FT/NC: fresh embryo transfer/naturally-conceived. VT/NC: vitrified embryo transfer/naturally-conceived. VT/FT: vitrified embryo transfer/ fresh embryo transfer. FC: fold change.

Metabolic Pathway Metabolite	FT/NC			VT/NC			VT/FT		
	FC	*p*-Value	adj *p*-Value	FC	*p*-Value	adj *p*-Value	FC	*p*-Value	adj *p*-Value
**Krebs cycle**									
Succinate	−0.86 *	0.03	0.37	−1.13 *	0.02	0.20	−0.27	0.62	0.92
**Biosynthesis of amino acids**									
Glutamine	−0.99 *	0.05	0.37	−1.43 *	0.03	0.20	−0.43	0.60	0.92
Threonine	−1.19 *	0.04	0.37	−1.10 *	0.05	0.27	0.08	0.92	0.92
Proline	−0.20	0.31	0.81	−0.54 *	0.04	0.20	−0.34	0.17	0.92
Valine	0.35	0.61	0.96	1.30 *	0.03	0.20	0.95 *	0.05	0.92
**Biosynthesis of unsaturated fatty acids**									
Adrenic acid	−3.04 *	0.05	0.29	−4.90 *	0.05	0.23	−1.86	0.83	1.00
Arachidonic acid	−3.00 *	0.05	0.29	−4.43 *	0.05	0.23	−1.43	0.85	1.00
Docosapentaenoic acid	−2.23 *	0.05	0.29	−4.19 *	0.03	0.22	−1.96	0.67	1.00
Eicosapentaenoic acid	−1.45 *	0.05	0.29	−1.99 *	0.03	0.22	−0.54	0.70	1.00
**Arachidonic acid metabolism**									
Prostacyclin	−2.47 *	0.02	0.29	−3.14 *	0.02	0.22	−0.67	0.83	1.00
19-/20-hydroxyarachidonic acid	−1.83 *	0.04	0.29	−2.46 *	0.03	0.22	−0.63	0.75	1.00
11(12)oxido-5,8,14-eicosatrienoic acid	−1.83 *	0.04	0.29	−2.46 *	0.03	0.22	−0.63	0.75	1.00
14(15)oxido-5,8,11-eicosatrienoic acid	−1.83 *	0.04	0.29	−2.46 *	0.03	0.22	−0.63	0.75	1.00
5(6)oxido-8,11,14-eicosatrienoic acid	−1.90 *	0.04	0.29	−2.46 *	0.03	0.22	−0.56	0.78	1.00
5-HETE	−1.98 *	0.02	0.29	−2.95 *	0.01	0.22	−0.97	0.67	1.00

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
