# Peer review of "Metabolomic Analysis Reveals Changes in Preimplantation Embryos Following Fresh or Vitrified Transfer"

_ijms, 2020, doi:10.3390/ijms21197116_

Round 1

Reviewer 1 Report

Overall, the statistical analysis is up to standard. Results are given in an incomplete way and as such is difficult to judge the validity of the results.

Results are presented with no correction for multiple testing. Given the very limited sample size there is the risk that no significant difference will be found or that only a few metabolites will show significant different after correction. This lack is a very major shortcoming of the analysis which need to be addressed

The paper is exploratory at best and great care should be take to take for what results are: a possibly very weak indication which need to be validated in a larger study sample. This should be clearly discuss and indicated in the discussion/or in a section titled “limitations”. Results must be discussed with respect to the limited sample size.

Abstract

The results show that as in vitro manipulation was increased (NC<FT<VT) both embryo survival rate (0.91 ± 0.020, 0.78 ± 0.051 and 0.63 ± 0.054, for NC, FT and VT groups, respectively) and embryo size (3.2 ± 0.04 mm, 2.2 ± 0.07 mm, 1.7 ± 0.06

Please be consistent with the number of significant digits. The error (sd?) cannot have more significant digits (i.e. being more precise than the mean...  0.78 ± 0.051  MUST be 0.78 ± 0.05, 2.2 ± 0.07 Must be 2.2 ± 0.1 and so on

Results

Please correct significant digits. There all wrong, in the text and in the tables!

Please give exact p-values not <0.01

Line 87: Why? Means must be reported with the standard deviation (s.e.) NOT with the standard error of the mean (s.e.\sqrt(s.e.)) !! IT he standard error of the mean is a measure of the dispersion of sample means around the true population mean. But this not what must be reported. What must be reports is the deviation around the sample mean!! This is a really a bad misconception that cast doubts on authors understanding of basic statistical procedures.

Line 96. There is no such a thing like PCA diagrams. Those are called, score plot, loadings plots or biplot, depending on what is shown. In this case, a score plot

Loading plots or bi-plot should also presented and discussed. In this case probably a biplot would bear the most information.

Line 97 There is no such a thing like prediction ellipse. The author are referring to a 95% confidence ellipse. PCA is an exploratory method not an inferential one!!

Line 124 “Functional clustering of the metabolites revealed alterations in metabolic pathways related 124 to the Krebs cycle, amino acids metabolism, unsaturated fatty acid metabolism and arachidonic acid 125 metabolism”. How was this performed? Using some sort of enrichment test? What is the significant of the association? That is, how many metabolites per pathways respect to the full metabolic pathways have been identified to be changeing/altered?

Why 5 and 7 (foster) mother where used? Is this choice based on some sort of Power calculation that have not been reported in the paper?

Statistical analyses

Please write down the model used

Please report the full table or results, including the variance and significance associated with the random terms

Why P-values have not been corrected for multiple testing? Results must given with uncorrected and corrected p-values

I could not access the supplementary material, but a table is needed with mean and s.e. for each metabolite for each group.

Why data is not made publicly available? If there are reason this should be disclosed in the Manuscript. However, not making data available hampers knowledge advancement and make the study not reproducible and makes impossible to check results consistence and this, in 2020, is not acceptable anymore. Code for the statistical analysis should also be made available.

Author Response

Dear reviewer,

Here, a step-by-step response is offered to each of the points, addressing those modifications to it may contribute to the final publication of the manuscript. Remarks posited by the reviewers have been noted in bold, and our response has been offered in italics to facilitate its readability.

Overall, the statistical analysis is up to standard. Results are given in an incomplete way and as such is difficult to judge the validity of the results.

Results are presented with no correction for multiple testing. Given the very limited sample size there is the risk that no significant difference will be found or that only a few metabolites will show significant different after correction. This lack is a very major shortcoming of the analysis which need to be addressed.

Determining sample sizes in metabolomics is essential, but due to the complexity of these experiments, there are currently no standard methods for sample size estimation in metabolomics. In fact, statistical methods for estimating the false discovery rate (FDR) in metabolomics studies are not common. We agree that FDR offers more confidence than just p-value, but in conventional metabolomic approaches to multiple comparisons, correction is likely too stringent. Metabolomics data exhibit high collinearity among compounds on similar biochemical pathways; there is a greater likelihood for biological and statistical correlation.  The FDR procedure has been successfully applied to datasets with a relatively large number of true positive associations, such as gene expression data. The use of FDR procedures for metabolome studies is compounded by the correlated nature of individual metabolic pathways (and super- and sub-pathways, for that matter), and lack of available annotations for much of the metabolome. Despite, based on reviewer suggestion, we have implemented the FDR test (see statistical section and Supplementary Table 1). Indeed, few metabolites show significant different after correction. Related to this, the discussion has been maintained based on the p-value. Still, the fact of the lack of significance after the FDR test has been highlighted in a new section called limitations, as recommended by the reviewer. In this work, the selection of targeted metabolites is based on our previous studies, where we demonstrated that embryo manipulation induced long-term and transgenerational phenotypic, transcriptional and metabolic effects in rabbit born following vitrified embryo transfer.

The paper is exploratory at best and great care should be take to take for what results are: a possibly very weak indication which need to be validated in a larger study sample. This should be clearly discuss and indicated in the discussion/or in a section titled “limitations”. Results must be discussed with respect to the limited sample size.

It is worth mentioning that there is none study conducted with whole mammalian embryos to assess embryo metabolism after in vitro manipulation.  Due that little data exist about the effects of in vitro manipulation on embryo metabolism mammalian species, this study should have relevance for future studies. Despite those as mentioned above, and being aware of the limitations of the sample size, we agree on the need to add a section titled “limitations”. Further research is required to validate these results.

Abstract

The results show that as in vitro manipulation was increased (NC<FT<VT) both embryo survival rate (0.91 ± 0.020, 0.78 ± 0.051 and 0.63 ± 0.054, for NC, FT and VT groups, respectively) and embryo size (3.2 ± 0.04 mm, 2.2 ± 0.07 mm, 1.7 ± 0.06

It has been correct across the manuscript.

Please be consistent with the number of significant digits. The error (sd?) cannot have more significant digits (i.e. being more precise than the mean...  0.78 ± 0.051  MUST be 0.78 ± 0.05, 2.2 ± 0.07 Must be 2.2 ± 0.1 and so on

It has been correct across the manuscript.

Results

Please correct significant digits. There all wrong, in the text and in the tables!

It has been correct across the manuscript.

Please give exact p-values not <0.01

It has been included.

Line 87: Why? Means must be reported with the standard deviation (s.e.) NOT with the standard error of the mean (s.e.\sqrt(s.e.)) !! IT he standard error of the mean is a measure of the dispersion of sample means around the true population mean. But this not what must be reported. What must be reports is the deviation around the sample mean!! This is a really a bad misconception that cast doubts on authors understanding of basic statistical procedures.

All the results are expressed as standard deviation. This has been modified in the manuscript.  

Line 96. There is no such a thing like PCA diagrams. Those are called, score plot, loadings plots or biplot, depending on what is shown. In this case, a score plot

It has been changed.

Loading plots or bi-plot should also presented and discussed. In this case probably a biplot would bear the most information.

Principal component analysis biplot (PCA-biplot) of untargeted metabolomes has been created with BioVinci software (BioTuring Inc., San Diego, CA, USA)

Line 97 There is no such a thing like prediction ellipse. The author are referring to a 95% confidence ellipse. PCA is an exploratory method not an inferential one!!

Since the type of graph has been changed, it has been chosen not to incorporate any confidence ellipse

Line 124 “Functional clustering of the metabolites revealed alterations in metabolic pathways related 124 to the Krebs cycle, amino acids metabolism, unsaturated fatty acid metabolism and arachidonic acid 125 metabolism”. How was this performed? Using some sort of enrichment test? What is the significant of the association? That is, how many metabolites per pathways respect to the full metabolic pathways have been identified to be changeing/altered?

Known metabolites were investigated by building a database for target analysis based on the Kegg pathways and the metabolites involved in these pathways. If differential metabolites involved in a specific pathway were detected, alterations in that pathway are considered, but there is no statistical analysis involved. We fully understand the reviewer's planning. For example, DAVID software allows for enrichment analysis, but this software only runs with transcriptomics and proteomics data, not with metabolomics.

Why 5 and 7 (foster) mother where used? Is this choice based on some sort of Power calculation that have not been reported in the paper?

Based on our considerable experience, when we operate with higher than 100 embryos (from several donors and pooled embryos) per experimental group, and these pooled embryos are transferred to a minimum of 5 foster mothers, the donor and recipient effect are not significant. By mixing all embryos recovery from all the donor, and randomly choosing the embryo to transfer, this effect is discounted. In the case of the foster mother, and as it has been included in the present study (table 1), its effect is not significant. Furthermore, this consideration of sample size design also lies in the moral necessity of applying the 3R principle to reduce harm to animals.

Statistical analyses

Please write down the model used

The mixed linear model has been written as:

Yijk = μ + Pi + COj + eij,

where Yij was the trait to analyse, µ was the general mean, Pi was the fixed effect of experimental groups (VT, FT and NC), COj was the random effect of donor mother and eij was the error term of the model.

Please report the full table or results, including the variance and significance associated with the random terms

It has been included

Why P-values have not been corrected for multiple testing? Results must given with uncorrected and corrected p-values

It has been implemented

I could not access the supplementary material, but a table is needed with mean and s.e. for each metabolite for each group.

Supplementary material has been completed

Why data is not made publicly available? If there are reason this should be disclosed in the Manuscript. However, not making data available hampers knowledge advancement and make the study not reproducible and makes impossible to check results consistence and this, in 2020, is not acceptable anymore. Code for the statistical analysis should also be made available.

It was a manuscript submission error. In my career, all the information is always available in repositories (when it is possible), or at the researcher demand. To date, there are no repositories for metabolic data.

Reviewer 2 Report

The article is well organised like the project of the study. The demonstration of injuries to the embryos by manipulating it is strong and supported by an elegant data’s set.

Author Response

Dear reviewer, 

Thank you very much for your words and support in publishing this study.

Round 2

Reviewer 1 Report

The authors have addressed most of my comments.

However, corrected p-value should be given in the main text not as supplementary material, since correction for multiple testing is a fundamental step of statistical analysis, not a supplementary step. Please add corrected p-values to Table 2 and in all other places so that a reader can do an informed decision on the significance of the results.I agree the correction increases the risk of false positives, but which risk to take can and should be left to the reader.

When giving p-values in the text please state both corrected and uncorrected i.e (P-value = xxxx, Ajusted P-value = yyy).

Line 100

however, no significant associations were observed(FDR <0.05).

If not association was observed I would expect FDR > 0.05

Line 102

Unsupervised PCA

PCA is unsupervised by definition...

Line 127

Principal component analysis biplots

it is Principal componentS analysis

Line 133

The T-test analysis

The t-test

Line 327

Yij = μ + Pi + COj + eij

Please use a decent mathemtical notation

Yij = μ + Pi + COj + eij

and state what the subscript i and j refer to

Line 331

The random effect donor mother (COj) had 7 levels.

if it is random how can have it 7 levels? Please rephrase

Line 349 and following

Please do not mix experiments and statistical methods, move this paragraph under statistical analysis.

Also make clear sub-paragraphs for each step of the experimental procedure

Line 350

Please reference the BH method

Abbreviations:

PCA-biplot Principal component analysis biplot

Should be Principal componentS analysis biplot

Author Response

Here, a step-by-step response is offered to each of the points (all of them minor concerns) indicated by the reviewer, addressing those modifications to it may contribute to the final publication of the manuscript. Remarks posited by the reviewers have been noted in bold, and our response has been offered in italics to facilitate its readability.

The authors have addressed most of my comments.

However, corrected p-value should be given in the main text not as supplementary material, since correction for multiple testing is a fundamental step of statistical analysis, not a supplementary step. Please add corrected p-values to Table 2 and in all other places so that a reader can do an informed decision on the significance of the results.I agree the correction increases the risk of false positives, but which risk to take can and should be left to the reader.

When giving p-values in the text please state both corrected and uncorrected i.e (P-value = xxxx, Ajusted P-value = yyy).

 It has been modified

Line 100

however, no significant associations were observed(FDR <0.05).

If not association was observed I would expect FDR > 0.05

It has been modified

Line 102

Unsupervised PCA

PCA is unsupervised by definition...

 The term unsupervised has been removed

Line 127

Principal component analysis biplots

it is Principal componentS analysis

 It has been changed

Line 133

The T-test analysis

The t-test

It has been changed

Line 327

Yij = μ + Pi + COj + eij

Please use a decent mathemtical notation

Yij = μ + Pi + COj + eij

and state what the subscript i and j refer to

It has been changed

Line 331

The random effect donor mother (COj) had 7 levels. 
if it is random how can have it 7 levels? Please rephrase

It has been rephrase

Line 349 and following

Please do not mix experiments and statistical methods, move this paragraph under statistical analysis. Also make clear sub-paragraphs for each step of the experimental procedure

It has been changed

Line 350

Please reference the BH method

It has been included

Abbreviations:

PCA-biplot Principal component analysis biplot

Should be Principal componentS analysis biplot

 It has been changed